# Social support for breast cancer patients in the occupied Palestinian territory

**Mona I. A. Almuhtaseb, Francesca Alby** ✱*, **Cristina Zucchermaglio, Marilena Fatigante**

Department of Social and Developmental Psychology, Sapienza University of Rome, via dei Marsi, Rome, Italy

* francesca.alby@uniroma1.it

## Abstract

Previous research indicates that social support is beneficial to cancer patients in adjusting to the stress of the disease. Drawing on a qualitative content analysis of 36 semi-structured interviews, this article explores sources and types of social support in Arab-Palestinian women with breast cancer. Results show that members of the immediate family, husbands in particular, are reported to be the most supportive social sources. Given the limitations that characterize access to cancer care in the occupied Palestinian territory (OPT) and the collectivistic values of the society, women with breast cancer seem to rely mainly on their husbands to handle emotional, functional and informational needs. Emotional support includes the provision of care, trust, reassurance, and companionship. Functional support includes the practical assistance that the cancer patients receive in terms of financial support, attendance during treatment or help with domestic chores and childcare. Accessing appropriate informational support can be quite challenging in the OPT since available information is not always reliable. The family plays a key role in mediating communication with doctors. Contact with breast cancer patients and survivors is also a source of supporting information, with however a possible negative impact in terms of emotional coping. In this context, the immediate family becomes a fundamental resource for coping and a relational space that mediates connections with others, including doctors, acting as a "proxy" between the patient and the social environment. Findings are discussed in light of the historical and sociocultural context of the OPT.

## Introduction

Previous research indicates that social support is beneficial to cancer patients in adjusting to the stress of the disease [1]. Social relations have in fact proved to be important resources for patients to mobilize in coping with cancer [2]. Typically, such resources have a positive impact not only on the patient's psychological response to cancer, but also on her physiological reaction to the disease [3, 4].

There is a general agreement among researchers that disease usually restricts the patient from participating in social activities; this in turn means that a cancer patient's opportunities of interacting with others, and hence of accessing social support, may decrease. Alternatively,

**Data Availability Statement:** The complete data corpus that supports the findings of this study is not publicly available due to privacy reasons, since it contains information that could compromise

research participant privacy. However, all relevant data required to replicate the study's findings are within the paper.

**Funding:** This work was supported by the Sapienza University's research grant 2018 n. RP118164126B8AD1, P.I.: Francesca Alby.

**Competing interests:** The authors have declared that no competing interests exist.

patients themselves may decide to withdraw from their social relations network. In either case, this is related to the patient's experience of cancer which depends on such variables as the patient's demographic characteristics (such as age, gender, socioeconomic status), and medical condition (such as the site of malignancy, stage of disease, and type of treatment) [5].

In this regard, Kleinke [6] maintains that a person who has strong social support, viewed in terms of the number and strength of his relationships, is more likely to be optimistic and less likely to be depressed than those with fewer or weaker social relationships. In addition, social support increases the patients' feeling of reassurance and provides them with strength [7, 8].

Several studies focusing on social support for breast cancer patients have been conducted in high-income countries. For example, a Norwegian study conducted in 2011 utilizing inter-views with 21 breast cancer patients (41–73 years) who had already received diagnosis and were waiting for surgery, indicated that the period between diagnosis and surgery was charac-terized by a lot of fatigue and psychological stress. The results also revealed that the degree of support these patients were provided with, as well as the information and consultation they received from people who were close to them or from professionals and health care providers, reassured them. In addition, the participants reported that when they received the diagnosis results, they were accompanied by a close person [9]. On the other hand, Kroenke et al.'s study [2], which aimed to examine the impact of social relations on the lifestyle of 3,139 breast cancer patients in the USA, revealed that patients who were socially isolated had a low quality lifestyle and a deteriorating physical health, and social and emotional life. The results also indicated that large social networks and a coherent social support are associated with higher standards of living after the stage of diagnosis with breast cancer.

Al-Azri et al. [10] documented that in lower-income countries women with breast cancer are at an increased risk of physical and psychological morbidities after diagnosis, such as dis-tress, anxiety, depression and concerns relating to children and family burden, body image and sexual and marital relations [11, 12]. However, social barriers and the false beliefs about cancer and its association with death for many society members affect a patient's daily life even by avoiding mentioning its name or talking about topics related to the disease [12–21]. In addition, several studies have showed that in the Arab and Palestinian society, a powerful social stigma is still attached to cancer [13, 22, 23]. For this reason, breast cancer patients try to hide the diagnosis and treatment from their neighbors, friends and colleagues in the work-place. To do so, they generally refrain from attending social gatherings and participating in cel-ebrations [24–26]. Consequently, these patients are most likely reluctant to share their experience with others, or might be denied access to social support and may then become socially isolated for fear of disapproval or rejection [2, 27]. The obverse is also true; that is, increasing the size of a person's social network influences the amount of social support received [28].

One of the principles in health psychology is the hypothesis that social support provided by trustworthy people has a significant importance in confronting the challenges relating to a dis-ease (e.g. sharing bad news) [29]. This type of support can reduce the negative impacts on health [30]. In this respect, Kayser, Cheung, Rao, Chan, Chan, and Lo [31] analyzed breast can-cer dyadic coping strategies among a sample of couples from Asia (Hong Kong-China, India) and the United States. The analysis of the narratives of 28 couples revealed that there are four social and cultural factors which influence the process of coping with breast cancer: (1) family boundaries, (2) gender roles, (3) personal control, and (4) interdependence. Some couples in the study stated that they were able to overcome the prevailing cultural standards and conse-quently were able to achieve balance in their life and were capable of adapting positively with the disease. The study further indicated that there are clear differences between couples from Asian cultures and American couples in terms of involving the extended families of the couple

in each stage of the disease, including the discovery of the disease and treatment decisions (i.e. family members presented different suggestions). In the interviews, Chinese couples, for example, indicated that the degree of the disease impacted not only the couple but also the entire family. Similarly, Kayser, Watson, and Andrade [32], examined how couples describe their coping strategies with breast cancer and found that social relationships can advance or hinder the coping process when the wife has breast cancer. This can be attributed to what we call "the relationship awareness", which involves thinking about one's relationship in the context of the illness. In addition, the study indicated that the husband tends to concentrate more on problem-focused and task-oriented strategies, such as collecting information about the disease, particularly regarding treatment, while the patient (i.e. the wife) is more likely to be emotionally-focused and concerned about expressing her feelings and communicating her stress to her partner. A similar study [33] found that couples react as an emotional system rather than as individuals; the study also stressed the need to understand the factors that influence distress.

Based on the results of these studies, the critical role of social support in assisting individuals, especially breast cancer patients, should be emphasized. Social support provides cancer patients with care and attention, and helps them to overcome their fear and anxiety from the disease as well as to mitigate the difficulties they face during the various stages of the disease.

Research has shown that the family plays an indispensable role in providing social support. In the literature, however, the word "family" is used in different ways: sometimes it refers to the extended family; at other times, it is used in reference to the couple (i.e. husband and wife). The variable use of the word family is related to different social and cultural practices in the different countries [34–36]. Furthermore, the importance of the type of relationship varies by age such that being married is more important to younger individuals; whereas having a larger network of relatives and friends is more important as one grows older [27].

This article explores sources and types of perceived social support in Arab-Palestinian women with breast cancer. To date very few studies have focused on this particular population, where women are coping with cancer while living under occupation in difficult conditions.

Although the incidence rate of breast cancer is lower than that in Western countries or in Israel, in the Occupied Palestinian Territory, breast cancer is the second most common cause of death among women [24–26]. It is therefore particularly important to better understand women's coping resources through culturally sensitive studies that take into account the specificities of the context of the Occupied Palestinian Territory.

We aim to explore who they perceive to be helpful in facing their illness and in what ways. Specifically, we aim to answer the following questions: Who helps and supports these women during their illness? What are the types of social support that they receive?

## Materials and methods

### Methods

The study was conducted in two medical centers: 1) the Oncology Department at Beit Jala Government Hospital in Bethlehem Governorate, as it is a significant government hospital in the occupied Palestinian territory (OPT) which provides a large part of diagnostic and therapeutic services for oncology patients, and 2) Dunya Women's Cancer Center in Ramallah Governorate since it is the only non-profit center that provides early diagnostic services for breast cancer and gynaecological services in the occupied Palestinian territory.

The study was ethically approved by the Palestinian Ministry of Health. Contact with participants was mediated by the two medical centers. All participants were approached at the centers, informed of the aims of the study and requested to sign a consent form. Participants were also fully assured of the anonymity and confidentiality of all the information collected for

the purpose of the study. The data collection started on 18 January 2015 and ended on 15 April 2015 due to the increased tension in the area and the consequent restrictions to mobility enforced by the Israeli occupation.

The participants recruited for the study were 36 women selected according to the following inclusion criteria: 1) living in the occupied Palestinian territory; 2) been diagnosed with breast cancer between one month-three years prior to the interview; 3) having no previous history of other forms of cancer, since we were interested in how women cope with a "first time" cancer diagnosis and treatment; 4) providing verbal consent and signing an informed consent form.

In the current study, semi-structured interviews were conducted as a way to obtain "thick descriptions" of the women's experiences and perspectives [37, 38]. Narrative interviews with patients were selected as the main data gathering tool as "Telling stories about illness is [a means] to give a voice to the body" [39, p. 18].

The interview guide had three sections. The first section explored how the illness was discovered, the second section focused on the communication with doctors and the treatment decision-making process and the third section explored the coping resources used by the study participants, specifically the resources of support in facing cancer.

This article focuses on the answers given to questions within the third section (What was most helpful in coping with the disease? Who supported you in this experience? How did the others—family, relatives, friends—react to your illness?).

The interview also covered socio-demographic characteristics (age, place of residence, education, marital status, employment status, how many children, religion) and medical information (date of diagnosis, stage, type of treatment).

The interviews were conducted in a private room (in the hospital or at the center) by one of the authors (M.A.I.), who was an Arab-Palestinian woman, which facilitated communication with the interviewees and their involvement in the research (there was no refusal to participate). She presented herself as a researcher and a doctoral student in social psychology. The participants were informed about the aims of the study and signed a consent form. All interviews were conducted in the Palestinian dialect of Arabic. They were audio recorded, transcribed in Arabic and later translated into English.

The analytical procedures used on the data corpus included: 1) a verbatim transcription of all the interviews and 2) a qualitative content analysis of the interview transcripts. We used an inductive process for the construction of coding categories [40–42]. In a first phase, two researchers read the transcripts of the interviews identifying the categories of content to be used for coding. In a second phase, these categories were compared and subsequently used to carry out the coding of the transcripts. The analyzes were carried out independently by the two researchers. Doubtful cases were discussed with a third researcher until an agreement was reached. The "saturation" level of the data was a matter of discussion and the data collected were considered sufficient for a qualitative exploratory analysis of sources and types of social support. In the article the extracts selected: 1) were representative of the identified categories; 2) showed the variety of ways in which social support is realized within the identified categories.

## Results

### Participants' characteristics

The study participants included 36 Arab-Palestinian women diagnosed with breast cancer. Eight of the participants had a history of cancer in the family. The participants were residents of the OPT, and half of them were village residents. The participants' age ranged between 22–67 years. The educational level of participants ranged from total illiteracy to university level

education. The majority of the participants were married (N = 31); of the other participants, one was abandoned by her husband after she was diagnosed with cancer, one was widowed, one was divorced and three were single. Most of the interviewees were housewives (N = 24). The number of children for thirty participants ranged from 1 to 9 children, while only six participants had no children. Almost half the participants (N = 17) said that they did not know at which stage of the disease they were when they were diagnosed with cancer; five reported that they were diagnosed at stage I, nine participants in stage II, three in stage III, and three at stage IV. Regarding treatment, fourteen participants reported that they had completed treatment (i.e. chemotherapy), though they still had to undergo follow up tests and diagnostic images regularly according to post treatment follow up guidelines.

## Sources of social support for Arab-Palestinian breast cancer patients

Social relations were described by the interviewees as the main source of support in facing the illness, together with faith and religion [see also 43]. To be more specific, slightly less than half of the study participants (N = 17) asserted that their husbands were most supportive of them during their experience with the disease. Eleven participants said that their family, in the broad sense, stood beside them at all the stages of the disease. However, seven participants indicated that some of their family members were completely unaware of their disease. These participants were all women with children who wanted to keep their condition from their children in order to avoid causing them any pain or distress or anxiety about their illness, or to make sure that their school-age children remain focused on their study. Based on these results, it seems that certain members of the close family (husband, mother, siblings) may be selected to be involved in the care and support of a breast cancer patient, while other family members may not be involved or even informed about the disease (children, old parents) with the intent of protecting them from feelings of anxiety or distress. It is worth mentioning that several study participants reported receiving little or no social support from healthcare providers, social workers or friends.

Drawing both on the qualitative content analysis and on a review of the previous literature [7, 28], we identified three types of social support: emotional, functional and informational support. These results are presented in the following sections.

## Emotional support

Emotional support includes the provision of care, trust, reassurance, and companionship. The participants in this study described the powerful emotional support they had received from their husbands and from some other members of their close family at different stages of the illness (e.g. from their mothers or siblings). The support of these family members helped them keep their fears under control and continue taking care of other family members such as children.

During the interviews, some participants recalled the moment they talked with their husband, as reported by S. and G., or informed their parents, as reported by H., and received immediate emotional support (See Extracts 1,2 and 3).

**Extract 1.** S. (32 years old): *"My husband was always trying to be strong and supportive of me, and he never let me feel that I had cancer."*

**Extract 2.** G. (42 years old): *My husband has become closer to me and supported me a lot during my treatment period."*

**Extract 3.** H. (33 years old): *"When I got the diagnosis result, I directly went to my parents' house even though my specialist doctor had asked me not to tell my family about my condition so that it wouldn't affect me negatively, I couldn't avoid telling them. I feel comfortable when I see*

*my parents and I forget that I am ill, but when I am at home, I feel that I will die there and then*! *I feel much safer when I am with people!"*

This extract clearly shows the emotional conflict faced by women regarding hiding the cancer diagnosis from family members or communicating it to them. This conflict is present in the Palestinian society (and here voiced by the doctor) and related to fear of the social stigma associated with the cultural beliefs about cancer [13].

Another example of emotional support provided by a family member is illustrated by R., who was 20 years old when she was diagnosed with cancer and was helped by her brother (See Extract 4).

**Extract 4.** R. (22 years old): "*[My brother] was always with me during all the treatment stages. I used to share all my thoughts and feelings with him; he always made me stronger and supported me all the way through. I always had certain fears regarding the treatment and about the things that I used to hear, but he always advised me not to listen to anyone.*"

In the case of T. (42 years old), her mother was very supportive of her emotionally, because she had lived the same experience (See Extract 5).

**Extract 5.** T. (42 years old): "*My mother talks with me a lot to support me; as 12 years ago, she had cancer and survived.*"

In a different case, Sa., a 56-year old nurse, initially refused chemotherapy because she was concerned about the side effects, but later changed her mind thanks to the support and insistence of her husband to follow the treatment (See Extract 6).

**Extract 6.** Sa. (56 years old): "*In the beginning, I refused chemotherapy, but with the assistance of my husband, he convinced me.*"

Other interviews show the role played by family members in limiting loneliness and isolation for the participants. The majority of the participants reported that they were more inclined to "isolate" themselves from others mainly because they were afraid of the pity that some people may feel towards them and also of the many questions about their illness; this bothered them very much, especially after surgery and the start of chemotherapy. Husbands or friends helped them to leave such isolation, by encouraging them to get out of the house and communicate with people (See Extract 7 & 8).

**Extract 7.** M. (38 years old): "*My husband was very kind with me and tried to make me go out, especially when he noticed that I had started to sit alone and think about my situation.*"

**Extract 8.** G. (56 years old): "*I got away from people for two years and became isolated. Only a group of friends supported me through recreation programs and holding awareness workshops on the disease and treatment.*"

Some participants reported that interacting with people helped them to have a positive and optimistic view on life (See Extract 9).

**Extract 9.** Sn. (46 years old): *"I felt people's love for me and everybody was asking about me to make sure I was fine."*

Some participants, as in S.'s case (32 years old), reported that their families had tried to support them by advising them to "ignore the disease" and live as normally as possible, in accordance with a personal belief that this could help them overcome the difficulties that might occur in the course of the disease. Thus these families hid their feelings of anxiety and fear, and avoided talking about negative thoughts, to avoid any outpouring of sympathy and give them "strength"—especially during the treatment period. At the same time, some participants were able to hide their fears, anxieties, and despair from their family and tried to live as normally as possible (See Extract 10).

**Extract 10.** M. (38 years old): *"My family did not show any reaction in front of me; and always told me*: "*We hope that God will make you recover*" *and tried to support me all the time. And my children tried to deal with me in a normal way as if I was not sick.*"

This section provided evidence of the emotional support provided by husbands, close family members and friends in their attempt to provide the participants with emotional balance, reassurance, and assistance to avoid isolation and loneliness.

## Functional support

Functional support includes the practical assistance that the cancer patients receive in terms of financial support, attendance during treatment or help with domestic chores and childcare. The participants in this study reported that they had informed their families about having breast cancer, particularly since they needed someone to be with their children and take care of them.

An example of functional support is provided by A., a 37 year old working woman with a Master's degree and a mother of five with an advanced stage of the disease, who has been helped by her husband in domestic and family care (See Extract 11).

**Extract 11.** A. (37 years old):*"My family was very supportive of me in fighting this disease especially my husband, who took full responsibility of the house."*

S. (36 years old) decided to inform her sister, who was living in Jordan, about her condition, so that she could come to the OPT to take care of her children while she was receiving treatment (See Extract 12).

**Extract 12.** S. (36 years old): "*I told my sister who is living in Jordan to come here at the time of the surgery to sit with my children."*

In the present study, all the participants were usually accompanied by family members to treatment sessions or to medical visits (See Extract 13).

**Extract 13.** Sh. (32 years old): *"My husband was very cooperative and supportive of me and accompanied me to all tests and treatment sessions. . .he didn't leave me alone."*

In the occupied Palestinian territory it is a routine practice for at least one family member to accompany a patient during chemotherapy sessions, particularly because of the side effects which the patient may experience after each session, such as vomiting and fatigue.

Most participants reported receiving financial support from their families. In the majority of cases however, husbands in particular assumed all financial responsibilities. The participants, most of whom were housewives, were economically dependent either on their husband or a close family member.

In this study, one participant (B., 53 years old) described her husband's exceptionally extreme reaction to her condition. According to B., her husband abandoned her after he found out that she had been diagnosed with breast cancer. However, she was extremely grateful to her brother for supporting her financially and emotionally (See Extract 14).

**Extract 14.** B. (53 years old): "*My husband does not know anything about me, or where I get treatment and who is following-up my case closely until now. Since I found out I had cancer he has been telling me*: *"I don't believe you are ill".. . . this has been his reaction since the beginning. . . our marriage is over*! *He left me. . . he wants me to be just as I was before in the same image he still keeps in his mind. I see him only once a month and there is no communication between us*! (. . .) *My brother supported me financially and emotionally."*

Only one case of marriage breakdown due to cancer has been reported in the present study; however, similar cases have been documented in Arab countries [44].

Returning to the question of financial support, Ba., a 48 year-old single woman who lives with her family and is responsible for supporting her family financially, has been facing a difficult financial problem since the beginning of her cancer treatment, when she was regularly absent from her work, and her employer stopped paying her salary (See Extracts 15 & 16).

**Extract 15.** Ba. (48 years old): *"My colleagues at work supported me a lot. During my treatment period, the company where I work stopped my salary because I was absent from work."*

**Extract 16.** N. (49 years old): "*Friends supported me psychologically more than my family.*"

N., a single woman who lived alone in her own house, could not tell even her father that she had cancer because she was afraid that the news might affect his health. And her mother was dead. N. stated that she had received emotional and social support from her friends and this had helped her in facing the disease.

## Informational support

The study participants received supportive information from several sources: doctors, mainly oncologists and surgeons; family or relatives, especially if there was someone working in the medical field; breast cancer survivors and other cancer patients.

In this context, the participants indicated that they often asked for a second opinion before starting a treatment plan proposed by the doctor. They consulted other doctors to get more information about their health condition, and specific details about the therapeutic steps which had to be taken. These medical consultations were made both locally and abroad (e.g. in other Arab countries mainly Jordan, or in Europe or the US), especially if they had relatives or friends living outside the OPT (See Extract 17).

**Extract 17.** G. (42 years old): "*It is true that the doctor told me everything, but honestly it was not enough for me, I travelled to Jordan and consulted doctors at Al-Hussein Cancer; and what they told me was identical to what my doctor in the West Bank had told me.*"

It appears that the main reason for participants to seek outside information was their need to obtain clearer and more detailed information about their health condition and to confirm the treatment recommendation. In some cases the reason was to verify the surgeon's decision to perform a mastectomy.

However, not all the participants wanted to obtain the information by themselves. Some women asked their husbands to communicate with the doctors on their behalf, preferring not to hear anything related to the disease, as reported by S. (36 years old) who asked the doctor to deliver the diagnosis result to her husband as she did not want to receive it herself. Some of the participants asserted that this gave them "strength" as they received the support and assistance of their families (See Extracts 18, 19, 20).

**Extract 18.** S. (36 years old): "*I didn't want them to tell me over the phone, nor to ask me to go to the center to get the news. It was better for me that they communicate with my husband.*". . . "*So, my husband is trying to know everything regarding my case before I do, and he follows up on my health condition more than I do.*"

**Extract 19.** D. (67 years old): "*He (brother-in-law) simply told me that I had cancer and that my breast had to be removed! No other way would have been better to be informed about my condition since my brother-in-law is a brother to me.*"

**Extract 20.** *Sa. (56 years old)*: "*I was terrified by the result; my husband followed-up on everything and it was good that he informed me.*"

These data suggest that in the occupied Palestinian territory the family plays a key role in medical consultations and negotiates with the doctor regarding the manner in which the patient is informed, or not, about the diagnosis.

Other sources of information include internet searches to learn more about the disease in general or its treatment and side effects, and advice on meeting the nutritional needs of cancer patients, particularly during the chemotherapy period (See Extracts 21 & 22).

**Extract 21.** Y. (35 years old): "*[I read] medical brochures and I also listened to other people's experiences. I did not know anything about the disease and I needed now to know about its symptoms and treatment.*"

**Extract 22.** Nd. (49 years old): *"I use the internet to read about healthy food for cancer patients as well as the stages of treatment."*

In the following extracts we focus on the difficulties that cancer patients face in getting information from internet websites. Some websites, especially those in the Arabic language, contain inaccurate or outdated information. This makes finding accurate and up-to-date online information about cancer a great challenge for many patients and their families. Some participants reported receiving information from books or brochures, or from other patients. However, obtaining information is not always considered a good thing by participants; it might, in some cases, have negative consequences (see Extract 23).

**Extract 23.** L. (41 years old): *"I used to ask a lot of patients about their cases and my condition deteriorated; after that I decided to not ask anyone."*

L. clearly stated that after asking patients about their experience with the disease for some time, she stopped doing so as she found that it had a negative impact on her and increased her fear and anxiety about the disease symptoms and side effects of treatment and accompanying pain and weakness.

In this section we showed that accessing informational support can be quite challenging in the OPT since the information from websites in Arabic language is not always reliable and extra effort is required to double-check medical advice (e.g. by travelling abroad). The family also plays a key role in obtaining informational support, by mediating communication with doctors. Contact with breast cancer patients and survivors is an additional source of supporting information, with however a possible negative impact in terms of emotional coping [17]. Interviewees such as L. deliberately avoided interactions with other patients as a way to protect themselves from making parallels that might harm their hope and psychological well-being.

## Discussion

There is a lack of research conducted in the occupied Palestinian territory exploring how patients cope with cancer. While further studies are needed, this research can be seen as a contribution towards filling that gap by giving voice to a hardly considered population: Arab-Palestinian women with breast cancer. Given the nature of qualitative research, we do not aim to generalize results but to contribute to the situated understanding of cancer coping strategies within the local context of the occupied Palestinian territory.

Our study describes the significant role of social relations as the main resource for women with breast cancer to gain support and cope with their illness. Members of the close family—the husband in particular—are reported to be the most supportive social sources. The study documented the three main types of support (emotional, functional and informational) provided through social relations. Emotional support includes the provision of reassurance, balance, companionship; functional support consists of financial help, help with domestic work and childcare, practical assistance and attendance during treatments; informational support includes information about health conditions, treatments and side effects and nutritional advice.

In all three types of support, the close family, and the husband in particular, plays a key role. This is partly linked to the young age of the interviewees (older patients could, for example, have indicated their children as a source of support). In the literature on Arab countries, however, the role of the husband as a companion and caregiver of women with breast cancer seems to be rather controversial. Studies on breast cancer patients from eastern countries have reported cases of marriage breakdown [44–46], picturing the husband as a non-supportive source. In the present study, however, a marriage breakdown was documented in only one of 36 cases. Although caution must be taken in interpreting a qualitative study, this finding may

indicate a situation in the occupied Palestinian territory more similar to what has been found in non-Arab countries in the Middle East region such as Iran and Turkey [47, 48].

The centrality of the husband's support may be due to the restrictions imposed by the Israeli occupation on the mobility of Arab-Palestinian people and the existence of the Separation Wall [13, 49]. These difficulties obstruct access to both the health care system and the social support networks such as the extended family and other members of the nuclear family.

Every Arab-Palestinian citizen, who moves in the occupied Jerusalem or the West Bank, would go through military checkpoints for inspection. Going through these checkpoints may take several hours and this can sometimes lead to complications in their condition. Under these circumstances, a lot of people do not leave their homes unless in extreme necessity as they feel unsafe and are forced to wait for a long time at checkpoints [13].

In Western countries about 80–90% of patients are informed of the oncological diagnosis, while in other cultural contexts the percentages are much lower (between 0 and 50%) [50, 51].

In contrast to the literature focused on Western societies and culture, where there is more emphasis on individualistic frames of reference and more concern for individual autonomy, studies from the Middle East and East Asian contexts give greater emphasis to the role of the family and to collective decision-making that also influences the communicative practices of a cancer diagnosis [50].

## Conclusions

Given the limitations that characterize the access to health care in the OPT, Arab-Palestinian women rely on the immediate family to handle emotional, informational and practical needs, while respecting and perpetuating the cultural norm that encourages them to hide their illness from the social environment. In the occupied Palestinian territory, women with breast cancer therefore experience multiple difficulties: first, they live a disruptive experience because of the oncological illness; second, they find themselves in a historical and political context that makes access to cancer care particularly complicated; third, they are in a sociocultural context that stigmatizes cancer, encourages its concealment and promotes adherence to traditional customs. Studies on attitudes towards cancer among Arab people in the Middle East reveal prevalent fear, embarrassment, cultural barriers related to modesty and social stigma, and fatalistic beliefs regarding causes and outcomes of cancer [13, 24–26, 49]. In this context, the close family becomes a fundamental resource for coping and a relational space that in turn mediates connections with others, including doctors, acting as a "proxy" between the patient and the social environment. Given the nature of qualitative research, we do not aim to generalize results, but to contribute to the situated understanding of coping resources within the local context of the OPT, a unique situation of enduring conflict and insufficient healthcare. Further studies using different analytical approaches may verify the extension of these findings to the wider OPT population.

## Acknowledgments

The authors thank all of the patients who participated in this study and shared their experiences.

## Author Contributions

**Conceptualization:** Mona I. A. Almuhtaseb, Francesca Alby, Cristina Zucchermaglio.

**Data curation:** Mona I. A. Almuhtaseb, Francesca Alby.

**Methodology:** Francesca Alby, Marilena Fatigante.

**Writing – original draft:** Mona I. A. Almuhtaseb, Francesca Alby, Cristina Zucchermaglio, Marilena Fatigante.

**Writing – review & editing:** Francesca Alby, Marilena Fatigante.

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
