## [Decision Letter · Decision Letter 0]

8 Mar 2021

PONE-D-20-36979

Social support for breast cancer patients in the occupied Palestinian territory.

PLOS ONE

Dear Dr. Alby,

Thank you for submitting your manuscript to PLOS ONE. After careful consideration, we feel that it has merit but does not fully meet PLOS ONE’s publication criteria as it currently stands. Therefore, we invite you to submit a revised version of the manuscript that addresses the points raised during the review process.

Although Reviewer 1 recommends publication, and both reviewers gave positive comments on your manuscript, Reviewer 2 requested several clarifications about the interpretation of the data or how they support the conclusions.

We look forward to receiving your revised manuscript.

Sincerely,

Yann Benetreau, PhD

Senior Editor, *PLOS ONE*

Journal Requirements:

Additional Editor Comments (if provided):

Reviewers' comments:

Reviewer's Responses to Questions

**Comments to the Author**

1. Is the manuscript technically sound, and do the data support the conclusions?

Reviewer #1: Yes

Reviewer #2: Partly

2. Has the statistical analysis been performed appropriately and rigorously? 

Reviewer #1: Yes

Reviewer #2: N/A

3. Have the authors made all data underlying the findings in their manuscript fully available?

Reviewer #1: Yes

Reviewer #2: No

4. Is the manuscript presented in an intelligible fashion and written in standard English?

Reviewer #1: Yes

Reviewer #2: Yes

5. Review Comments to the Author

Reviewer #1: This is an excellent article!

It throws light on a Palestinian population living under occupation and difficult conditions. It gives information about how women cope with breast cancer under such circumstances, but also gives information about of how women in general cope with breast cancer. The results are therefore relevant also for other populations. - The analysis is well done, and the background (theory and previous literature) is relevant.

Regarding question 2 (above): (Statistical analysis), I answered "yes" as the qualitative analysis is good. There are, of course, no statistical analyses in a qualitative investigation.

Regarding question 3 (above): (Whether all data are available), - I also anwered "yes", as all relevant data seem to be analysed and sufficiently reported. Needless to say: one cannot, and should not, for ethical reasons, report absolutely all data in a qualitative report.

I suggest only these corrections: On page 17 a name, "Nadia" is mentionned. That should be omitted, and only the initial "N" should be used (as "N" is used for this informant a few sentences earlier in the manuscript. - In another place "Su" appears as what looks like a name. - That too should perhaps be shortened to just "S".

Reviewer #2: The article covers an important topic, however the paper needs considerable revision to be of publishable quality. One of the main (major) issues with the paper is that the authors do not adequately engage with the data presented through quotes; some statements are made without being adequately supported or explained. They may be true, but need more explanation and evidence from the authors, for example the issue of breast cancer being stigmatized. There also needs to be more thorough engagement with the literature in the discussion section. The authors compare the findings about husband support with others in the Arab world, but in their discussion there is an implicit assumption that 'Arab' culture is homogeneous, which is not the case, and cultures within any place are also not static.

More specific comments:

1) the transition from the introduction to the objectives could be improved by making the case as to why this study is important in Palestinian context

2) objectives should be in introduction rather than materials and methods; objectives also need to be written more clearly, especially first line

3) line 141 part about literature that 'gave voice' is a bit of an odd statement and perhaps better to say what methods they drew on

4) line 148, incorrect statement about Beit Jala, it is one of the major government hospitals in the West Bank providing oncology services but not the only one

5) line 158, not clear what is meant by security and travel as the justification. Is this for the researchers coming from abroad and limitations on their stay? Better to clarify

6) justification needs to be provided for inclusion criteria, especially 3 and 4. In local context, verbal consent is often deemed appropriate by IRB and ethical review committees, why did researchers insist on signed consent?

7) line 167 needs to be explained further

8) line 180-'adherence to research' odd wording

9) line 193, what was rendered sufficient?

10) line 215, not clear if according to literature or the women interviewed

11) line 227-228 where authors state 'attributed to young age and marital status"- it is not clear what is meant and this should be explained more, examples from the data would also be useful.

12) line 229, the statement makes it seem like all women stated they had no support from those outside family, but quotes later on contradict this (e.g. extracts 8 and 9)

13) for discussion of woman whose husband abandoned her, the quote on line 355 also states he comes once a month. It is a bit confusing.

14) beginning line 358: authors jump to loss of work, the transition is abrupt

15) paragraph beginning line 369, authors use N. and then Nadia, better to stick to one or the other and maybe initial for consistency, and ethical/privacy considerations.

16) same participant as above, authors state she didn't tell her family and then talk about support from friends. Were there family members who knew about her diagnosis and not support her? it is not clear

17) statement beginning on line 396 about family playing key role. This may be true, but authors should support through examples from study or references in the literature.

18) lines 437-439, authors talk about lack of reliable information. From whom? The internet or is this more specific to Palestinian context? It reads as if it is more specific to Palestinian context, but at the same time it is not really explained. The references to unreliable information in Arabic on the internet would not limit this issue to Palestinian context.

19) line 442, what negative impact? Explain.

20) line 468, not clear how explanation going back to political context explains centrality of husband as support provider. It is not clear and not convincing from the argument. Also, why should we assume that husbands won't be supportive?

21) statement that breast cancer is stigmatized needs to be explained further.

22) for ethical considerations, were patients also assured that their participation (or not) would not affect their access to treatment?

23) Authors used an inductive approach, did this impact what kinds of support were identified? Were there other forms of support that were not identified during this approach? Important to discuss limitations of analytical approach as well

24) the manuscript can use some copy editing, some use of language reads as a bit odd.

6. PLOS authors have the option to publish the peer review history of their article (what does this mean?). If published, this will include your full peer review and any attached files.

Reviewer #1: **Yes: **Torill Christine Lindstrøm

Reviewer #2: No

---

## [Author Response · Author response to Decision Letter 0]

17 Mar 2021

Response to Reviewers

In what follows we outline the main changes that we have made to our manuscript by answering to each of the reviewers’ requests.

As suggested, we have tracked the changes made to our original submission. We provide also another file that shows the changes made by an English proofreader. We also provide a final, clean version. 

Reviewer #1: This is an excellent article!

It throws light on a Palestinian population living under occupation and difficult conditions. It gives information about how women cope with breast cancer under such circumstances, but also gives information about of how women in general cope with breast cancer. The results are therefore relevant also for other populations. - The analysis is well done, and the background (theory and previous literature) is relevant.

Regarding question 2 (above): (Statistical analysis), I answered "yes" as the qualitative analysis is good. There are, of course, no statistical analyses in a qualitative investigation.

Regarding question 3 (above): (Whether all data are available), - I also answered "yes", as all relevant data seem to be analysed and sufficiently reported. Needless to say: one cannot, and should not, for ethical reasons, report absolutely all data in a qualitative report.

I suggest only these corrections: On page 17 a name, "Nadia" is mentioned. That should be omitted, and only the initial "N" should be used (as "N" is used for this informant a few sentences earlier in the manuscript. - In another place "Su" appears as what looks like a name. - That too should perhaps be shortened to just "S".

Changed as suggested.

Reviewer #2: The article covers an important topic, however the paper needs considerable revision to be of publishable quality. One of the main (major) issues with the paper is that the authors do not adequately engage with the data presented through quotes; some statements are made without being adequately supported or explained. They may be true, but need more explanation and evidence from the authors, for example the issue of breast cancer being stigmatized. There also needs to be more thorough engagement with the literature in the discussion section. The authors compare the findings about husband support with others in the Arab world, but in their discussion there is an implicit assumption that 'Arab' culture is homogeneous, which is not the case, and cultures within any place are also not static.

More specific comments:

1) the transition from the introduction to the objectives could be improved by making the case as to why this study is important in Palestinian context

2) objectives should be in introduction rather than materials and methods; objectives also need to be written more clearly, especially first line

3) line 141 part about literature that 'gave voice' is a bit of an odd statement and perhaps better to say what methods they drew on

1-2-3: Objectives have been moved in the introduction and partially reformulated. In order to making a case for why the study is important, we added few lines about cancer incidence in the Palestinian territory and the relevance of studying coping strategies under these local difficult circumstances (being under occupation) . In what used to be line 141 “gave voice” has been erased and the line rephrased.

Reference to the importance of local culture (addressed by Rev2 at the beginning of this letter) and culturally sensitive studies has been added at the end of the introduction.

4) line 148, incorrect statement about Beit Jala, it is one of the major government hospitals in the West Bank providing oncology services but not the only one

Correction made

5) line 158, not clear what is meant by security and travel as the justification. Is this for the researchers coming from abroad and limitations on their stay? Better to clarify

Clarification added

6) justification needs to be provided for inclusion criteria, especially 3 and 4. In local context, verbal consent is often deemed appropriate by IRB and ethical review committees, why did researchers insist on signed consent?

Few lines of explanation have been added. Verbal consent was asked and followed by a written consent that also provided the opportunity to give the patients written information about the research and researchers’ contacts

7) line 167 needs to be explained further

This line has been eliminated

8) line 180-'adherence to research' odd wording

Rephrased

9) line 193, what was rendered sufficient?

Clarification added

10) line 215, not clear if according to literature or the women interviewed

Clarification added

11) line 227-228 where authors state 'attributed to young age and marital status"- it is not clear what is meant and this should be explained more, examples from the data would also be useful.

This statement has been rephrased and moved later on in the discussion section. 

12) line 229, the statement makes it seem like all women stated they had no support from those outside family, but quotes later on contradict this (e.g. extracts 8 and 9)

Rephrased 

13) for discussion of woman whose husband abandoned her, the quote on line 355 also states he comes once a month. It is a bit confusing.

The interviewee states clearly that their marriage is finished. She still meets the husband occasionally but without having the feeling of a real communication between them

14) beginning line 358: authors jump to loss of work, the transition is abrupt

A line that reconnects with the financial support topic has been added

15) paragraph beginning line 369, authors use N. and then Nadia, better to stick to one or the other and maybe initial for consistency, and ethical/privacy considerations.

Changed in N.

16) same participant as above, authors state she didn't tell her family and then talk about support from friends. Were there family members who knew about her diagnosis and not support her? it is not clear

Rephrased. (She did not tell her diagnosis to the family).

17) statement beginning on line 396 about family playing key role. This may be true, but authors should support through examples from study or references in the literature.

The statement has been moved after extracts 18,19,20 that provide evidence for it

18) lines 437-439, authors talk about lack of reliable information. From whom? The internet or is this more specific to Palestinian context? It reads as if it is more specific to Palestinian context, but at the same time it is not really explained. The references to unreliable information in Arabic on the internet would not limit this issue to Palestinian context.

Specification added

19) line 442, what negative impact? Explain.

Few lines of explanation have been added

20) line 468, not clear how explanation going back to political context explains centrality of husband as support provider. It is not clear and not convincing from the argument. Also, why should we assume that husbands won't be supportive?

Explanation added (socio-political restriction to mobility obstruct access to both the health care system and the social support networks such as the extended family and other members of the nuclear family). A previous paragraph of the discussion section describes how in the literature on Arab countries the role of the husband as a companion and caregiver of women with breast cancer is controversial, picturing often the husband as a non supporting source.

21) statement that breast cancer is stigmatized needs to be explained further.

Main results of the studies quoted on this topic are reported.

22) for ethical considerations, were patients also assured that their participation (or not) would not affect their access to treatment?

Yes. Moreover, several patients already completed treatments.

23) Authors used an inductive approach, did this impact what kinds of support were identified? Were there other forms of support that were not identified during this approach? Important to discuss limitations of analytical approach as well

Limitations and reference to other analytical approaches have been added

24) the manuscript can use some copy editing, some use of language reads as a bit odd.

The manuscript has been proofread by a fluent English speaker and changes have been made.

---

## [Editor Report · Decision Letter 1]

14 Apr 2021

PONE-D-20-36979R1

Social support for breast cancer patients in the occupied Palestinian territory.

PLOS ONE

Dear Dr. Alby,

Thank you for submitting your manuscript to PLOS ONE. After careful consideration, we feel that it has merit but does not fully meet PLOS ONE’s publication criteria as it currently stands. Therefore, we invite you to submit a revised version of the manuscript that addresses the points raised during the review process.

We look forward to receiving your revised manuscript.

Kind regards,

Weeam Hammoudeh

Academic Editor

PLOS ONE

Journal Requirements:

Additional Editor Comments (if provided):

The authors have revised and improved the manuscript substantially. There are a few things in the text. The following sentence is not very clear. Framing the issue of willingness to go doesn't fully reflect the reality. Also rather than saying 'should go through', 'would go through' is more suitable, since there is a value judgement in should, and I doubt what the others are trying to say is that this should be the case but rather that it is likely to be the case.

478 Every Arab-Palestinian citizen, who is willing to go to the occupied Jerusalem or move to

479 the West Bank, should go through military checkpoints for inspection. Going through these

480 checkpoints may take several hours and this can sometimes lead to complications in their

481 condition.

When referring to 'western individualistic culture', it presents as if there is a homogenous culture, and while individualistic inclinations may be more common, maybe you could say something like where there is more emphasis on individualistic frames of reference and more concern for individual autonomy in the literature focused on Western societies and culture, studies from the Middle East and East Asian contexts give greater emphasis to the role of the family and to collective decision-making that also influences the communicative practices of a cancer diagnosis. (this is just a suggestion, and obviously alternative wording can be posed that would allow for some of that nuance to come through).

---

## [Author Response · Author response to Decision Letter 1]

10 May 2021

Dear PLOS ONE Editor and anonymous reviewers,

My co-authors and I are grateful for the attention given to our manuscript PONE-D-20-3697R1

and the suggestions offered by the editor. We changed the manuscript accordingly in lines 478-481 and rephrase the reference to western individualistic culture as recommended.

Kind regards,

The Authors

---

## [Editor Report · Decision Letter 2]

19 May 2021

Social support for breast cancer patients in the occupied Palestinian territory.

PONE-D-20-36979R2

Dear Dr. Alby,

We’re pleased to inform you that your manuscript has been judged scientifically suitable for publication and will be formally accepted for publication once it meets all outstanding technical requirements.

Kind regards,

Weeam Hammoudeh

Guest Editor

PLOS ONE
---

## [Editor Report · Acceptance letter]

24 May 2021

PONE-D-20-36979R2 

Social support for breast cancer patients in the occupied Palestinian territory. 

Dear Dr. Alby:

I'm pleased to inform you that your manuscript has been deemed suitable for publication in PLOS ONE. Congratulations! Your manuscript is now with our production department. 

Kind regards, 

on behalf of

Dr. Weeam Hammoudeh 

Guest Editor

PLOS ONE